# Factor analysis, sparse PCA, and Sum of Ranking Differences-based improvements of the Promethee-GAIA multicriteria decision support technique

János Abonyi[1]*, Tímea Czvetkó[1], Zsolt T. Kosztyán[2], Károly Héberger[3]

**1** MTA-PE "Lendület" Complex Systems Monitoring Research Group, University of Pannonia, Veszprém, Hungary, **2** Department of Quantitative Methods, Faculty of Business and Economics, University of Pannonia, Veszprém, Hungary, **3** ELKH Research Centre for Natural Sciences, Institute of Excellence of the Hungarian Academy of Sciences, Budapest, Hungary

* janos@abonyilab.com

**Data Availability Statement:** All relevant data are within the manuscript and its Supporting information files.

## Abstract

The Promethee-GAIA method is a multicriteria decision support technique that defines the aggregated ranks of multiple criteria and visualizes them based on Principal Component Analysis (PCA). In the case of numerous criteria, the PCA biplot-based visualization do not perceive how a criterion influences the decision problem. The central question is how the Promethee-GAIA-based decision-making process can be improved to gain more interpretable results that reveal more characteristic inner relationships between the criteria. To improve the Promethee-GAIA method, we suggest three techniques that eliminate redundant criteria as well as clearly outline, which criterion belongs to which factor and explore the similarities between criteria. These methods are the following: A) Principal factoring with rotation and communality analysis (P-PFA), B) the integration of Sparse PCA into the Promethee II method (P-sPCA), and C) the Sum of Ranking Differences method (P-SRD). The suggested methods are presented through an I4.0+ dataset that measures the Industry 4.0 readiness of NUTS 2-classified regions. The proposed methods are useful tools for handling multicriteria ranking problems, if the number of criteria is numerous.

## Introduction

The primary goal of this paper is to support Promethee-based decision-making by integrating methods that enable relationships between criteria to be better identified and improve its interpretability.

Promethee I partial ranking and Promethee II complete ranking methods were first introduced by J. P. Brans in 1982 [1]. In this study, only the Promethee II method is considered (hereinafter referred to as Promethee). This method defines preference functions associated with each criterion that can be aggregated and weighted to identify the relative importance of a variable.

**Funding:** This work was supported by the TKP2020-NKA-10 project financed under the 2020-4.1.1-TKP2020 Thematic Excellence Programme by the National Research, Development and Innovation Fund of Hungary. The contribution of Károly Héberger, project no. OTKA 134260, was supported by the Ministry of Innovation and Technology of Hungary from the National Research, Development and Innovation Fund, financed under the K type funding scheme.

**Competing interests:** The authors have declared that no competing interests exist.

**Abbreviations: Latin Symbols:** *A*, set of alternatives; *d*, absolute value of differences between the reference ranking and individual ones; *e*, unique factor; *F*, standardized factor score matrix; *G*, set of criteria; $h^2$, common variance; *m*, number of alternatives; *M*, matrix of criteria net flows; *n*, number of criteria; $P(a_i, a_j)$, degree of preference shows how alternative $a_i$ is preferred instead of alternative $a_j$; *P*, matrix of loadings; *q*, partial correlation coefficient; *r*, Pearson product-moment correlation coefficient; *U*, matrix of unique variances; *Z*, standardized original data matrix; **Greek Symbols:** *δ*, percentage of total information preserved; *π*, aggregated preference index; $\phi^-$, negative outranking flow; *ϕ*, net outranking flow; $\phi^+$, positive outranking flow; *ω*, weights of criteria; **Abbreviations:** *AHP*, Analytic hierarchy process; *ANOV*, A Analysis of Variance; *ANP*, Analytic Network Process; *CRRN*, Comparison of Ranks with Random Numbers; *DEA*, Data envelopment analysis; *GAIA*, Geometrical Analysis for Interactive Aid; *GRA*, Grey Relational Analysis; *KMO*, Kaiser-Meyer-Olkin method; *MCDM*, Multicriteria Decision Making (also referred to as MCD analysis, (MDCA) or multiobjective optimization (MOO), Post-Pareto optimization (PPO) etc.); *MODRO*, Multiobjective discrete robust optimization; *MSA*, Measure of Sampling Adequacy; *P-GAIA*, Promethee II—GAIA method; *P-PFA*, Coupling of the Promethee-GAIA method with principal factoring with rotation and communality analysis; *P-sPCA*, Coupling of the Promethee-GAIA method with Sparse Principal Component Analysis; *P-SRD*, Coupling of the Promethee-GAIA method with the Sum of Ranking Differences method; *PC*, Principal Component; *PCA*, Principal Component Analysis; *PFN*, Pythagorean fuzzy numbers; *sPCA*, Sparse Principal Component Analysis; *SRD*, Sum of Ranking Differences; *TOPSIS*, Technique for Order Preference by Similarity to Ideal Solution.

Although the application of Promethee is generally used for multicriteria decision aid, several fields of application are covered [2]. This method is used in finance [3], education [4], health care, circular economy [5] black, logistics and transportation [6], hydrology and water management [7] as well as manufacturing and assembly [8, 9]. It is present in the environmental field such as in aiding strategic planning for the mitigation of climate change [10], ranking municipal solid waste treatment alternatives [11], prioritizing environmental projects [12] or integrating Promethee into automated multi-material approaches [13]. In terms of socio-economic and corporate aspects, research has been conducted to support construction project management [14], innovation management [15], the selection of outsourcing functions for human resource management [16], the selection of suppliers based on their corporate social responsibility practices [17] and the analysis of the link between corporate governance and firm performance [18].

In 1988, J. P. Brans and B. Mareschal introduced a visual interactive module named GAIA, which provides a graphical representation to support the Promethee methodology [19]. The GAIA method promotes the interpretability of the results by providing visual assistance to understand the conflicting aspects of the criteria and tackle the problem concerning the weights associated with them [19]. The Promethee-GAIA methodology is the visual interpretation of the generated matrix of criterion net flows that relies on Principal Component Analysis (PCA) to project information [20]. PCA is a well-known method for data processing and dimensionality reduction by taking the linear combination of criteria. The biplot of PCA shows the principal components, and the loading vectors of alternatives [21]. The GAIA plane only includes a percentage *δ* of the total information. The "visual representation of the main characteristics of the decision problem, such as conflicts existing between criteria and specific profiles of the actions" is provided for the decision-maker [22]. The length of a criterion axis in the GAIA plane shows how to discriminate a criterion, while the orientation of the axes indicates how similar the preferences of criteria are. Furthermore, the location of points representing the alternatives provides information about similarities between them and how good the alternatives are on a particular criterion [19, 23]. The drawback of PCA is that in the case of numerous criteria, the interpretation of principal components is often difficult. The linear combination of criteria and the loadings are typically non-zero, which makes it difficult to clearly perceive the importance of criteria [24]. All the original criteria determine the principal components and summarize the information in a few factors. The factors themselves are still constructed using all the criteria [25]. Rotation techniques are commonly used methods to support the interpretation of the principal components [26]. Furthermore, alternatives to PCA have been proposed, which identify sparse and potentially interpretable factors, such as the sparse PCA. It creates modified PCs with sparse loadings based on the fact that PCA can be written as a regression-type optimization problem, thus the elastic net can be directly integrated into the regression criterion [24]. It is essential to decide and clearly understand which criterion belongs to which factor.

Some key sources are interpreted here from the plethora of publications coupling both MCDM and PCA mainly for sustainable development topics.

The common visualization way, the PCA biplot was rediscovered for MCDM examples [27]. Only Promethee and PCA was coupled, in our terminology the Promethee-GAIA, similarities and clustering was revealed for criteria and alternatives alike. Pavan and Todechini's different terminology also covers the problematic: "contradictions in the ranking are bound to exist and the higher the number of criteria, the higher the probability that contradictions in the ranking occur. . ." [28, p. 167]. They have defined a lot of indices, which allow the comparison of ranked sets [28]. Composite sustainability indicators were specified using data

envelopment analysis (DEA) integrated with MCDM methods for ranking of farm in Campos County (Spain) [29]. Their approach was clearly differentiated from an earlier one based on PCA [29].

PCA was also used to calculate attribute weights after rotation. First the correlation structure was examined, then, Kaiser-Meyer-Olkin measure and squared factor loadings calculated for obtaining weights [30]. However, Randjelović *et al.* 's approach are suitable for highly correlated matrices and have not been tested in MCDA environment [30].

According to Dugger *et al.*, personal selection process was aimed to be modernized using PCA to determine weights for multicriteria decision [31]. Davoudabadi *et al.*, selected suppliers by integrating PCA and DEA; four decision making scenario was compared using weights. Their proposed method gives totally different rankings as the first- and last aggregation for weights' importance determination are different [32].

The ambiguity of the MCDM methods was aimed to be eliminated by defining a new closeness index for Pythagorean fuzzy numbers (PFNs) [33]. Zhang also preluded a ranking system based on PFNs without any PCA or factor analysis [33]. Conflicting objectives were converting into one to unify cost functions: Grey Relational Analysis (GRA), was associated with PCA on the example of automobile industry (design). Sun *et al.* called their approach MODRO (multi-objective discrete robust optimization) [34].

Six alternatives were compared pairwise with integrated AHP–TOPSIS (analytic hierarchy process (AHP) and Technique for Order Preference by Similarity to Ideal Solution (TOPSIS)) algorithm in weighted and unweighted forms for selection of electric buses. The visualization embraces performance indicators: radar plots and line plots of scenarios without any application of PCA [35].

Eleven publications about MCDM methods proved to have serious deficiencies [36] including high computational complexity (and hence high time demand of calculations), inconsistency and problems of pairwise comparisons. Kumar *et al.* defined twelve quality of criteria and weights were calculated by TOPSIS (a best worst method) They claim a framework for the application of one MCDM method to "trustworthy" cloud service selection without PCA or any variants of factor analysis.

Two fuzzy hybrid multicriteria decision-making (MCDM) models were developed because fuzzy sets can cope uncertain and ambiguous character of resilience engineering. very well, weights of resilience indicators were calculated as well as pairwise consistency check by eigenvalue analysis [37].

A characteristic picture is outlined by Bortoluzzi *et al.* [38]. They undertook a bibliometric analysis using 142 papers and aggregated key performance indicators (KPI), form the word cloud (KPIs are termed as criteria in this work). The renewable energy technologies were clustered according to performance indicators and by MCDM models. Two or three clusters could be observed on 2-dimensional PCA scatter plots. Variance analysis (ANOVA) was carried out on maximal nine MCDM techniques. In some cases, the "Analytic Network Process (ANP)" and "PROMETHEE" models were discarded from the sample because they did not present the required level of significance [38].

Numerical cardinal comprehensive evaluation was complemented the GAIA visualization of PROMETHEE. The fuzzy preferences of the Borda score are, in fact, the reversed sum data fusion. Inclusive development index was also suggested to measure welfare of the nations, instead of the gross national product. A biplot of GAIA and sigma-mue ($\sigma - \mu$) planes was constructed to visualize the nations and the performance indicators [39].

It can be traced out form the above short summary of literature tendencies that most methods cannot handle the deficiencies or only one of them: Essential shortcomings ere enumerated here, i) the direction of ranking might be dubious, and not handled; ii) the large number

of alternatives makes the interpretation almost impossible; iii) many algorithms delivered to select (subjective) weights; iv) they apply pairwise comparisons, such ignoring higher degree of interactions; v) frameworks are often claimed, but it generally means a flowchart for a special case study, vi) All of the above deficiencies may cause inconsistent results.

Thereby, the aim of our study is to fill the gaps mentioned above and improve Promethee-GAIA to select relevant indicators, which can be attached to a unique factor unambiguously. This aim is fulfilled by integrating the following techniques into a framework of principal factor analysis, sparse PCA and SRD (the latter can cope with rank reversal problem). Our study I4.0+ dataset includes 314 alternatives (NUTS 2 classified European regions) and 69 criteria categorized into five dimensions, all other technique applied much less criteria and/or alternatives. Further case studies can be found in the discussion section.

The contributions of the work are as follows:

- Supports subjective multicriteria decision-making in terms of large number of criteria.

- Improve Promethee-based decision-making processes to gain more interpretable results.

- The relationships between criteria can be better characterised.

- Regional rank of Industry 4.0 readiness is analysed.

The paper shows how the Promethee-based decision-making process can be improved to gain more interpretable results that reveal more characteristic relationships between criteria. In the following, three methods are suggested to be integrated into the Promethee-based decision-making process that increases interpretability by eliminating redundant criteria from the formation of principal components as well as identifying relevant groups of criterion and analyzing similarities between criteria.

The Materials and Methods section introduces the proposed multicriteria decision support framework, then discuss the Promethee method, the principal factoring with rotations and communality analysis (P-PFA), then the approach of integrating sparse PCA (P-sPCA) and the SRD (P-SRD) method. Finally, in the Results and Discussion Section, the proposed concepts of developing the Promethee-GAIA methodology are applied and interpreted through a detailed case study related to the analysis of regional Industry 4.0 (I4.0+) readiness of economic regions.

## Materials and methods

### The proposed multicriteria decision support framework

The scheme of the methods is shown in Fig 1. The input matrix of each method is the Promethee-defined matrix of criterion net flows, the net flows of which include the degrees of preference given by the criteria [40]. The original Promethee uses the GAIA biplot to identify relationships between the criteria, while the suggested methods approach this problem from different perspectives as discussed below:

Method A) **P-PFA**: Principal factoring with rotation and communality analysis to efficiently select criteria and better interpret results.

The P-PFA method improves Promethee by carefully selecting and eliminating redundant (multicollinear), irrelevant and common criteria, thereby improving the factor structure for the purpose of better interpretation. Furthermore, it aims to clearly define groups of criteria involved in factors. The Promethee-GAIA uses PCA to interpret the results in two-dimensional space. PCA summarises the original criteria into components and assumes that the common variance is maximized and no unique variance is present in each criterion [41]. In

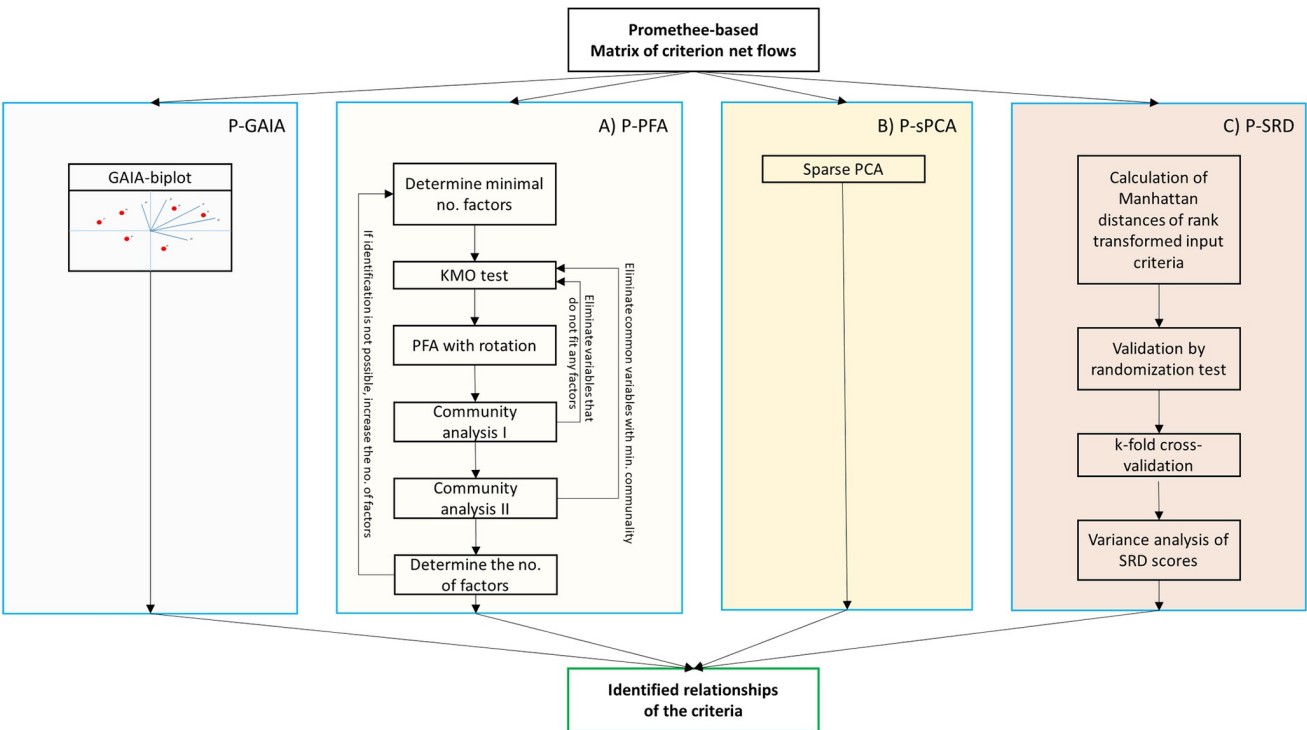

**Fig 1. Promethee-based decision-making improvement using the following methods: Method A) P-PFA—Principal factoring with rotation and communality analysis; Method B) P-sPCA—Integration of sparse PCA into the Promethee methodology; Method C) P-SRD—Sum of Ranking Differences to qualify the consistency of the criteria.**

contrast, principal factoring assumes a substantial amount of unique variance and reliable, common variance. Principal factoring seeks to determine the minimum number of factors which can account for the common variance of a set of criteria. P-PFA drops overly multi-collinear criteria based on the Measure of Sampling Adequacy (MSA) value that identifies how suitable the data is for factor analysis [42]. It predicts, if data are likely to factor well based on correlation and partial correlation. Irrelevant criteria are dropped, which are considered not to fit any factors because of their low level of communality. Subsequently, criteria with the lowest communality values are eliminated from the relevant criteria. The P-PFA method groups criteria into factors while accounting for a maximum amount of variance of observed criteria. In order to support interpretation, varimax rotation is used to avoid most of the criteria belonging to the first factor [43].

Method B) **P-sPCA**: The integration of sparse PCA for the purpose of improving the interpretability of the results.

The P-sPCA method improves Promethee by combining the advantages of PCA with the formulation of elastic net regression formulation to create sparse loading vectors [24]. The primary aim of P-sPCA is to support decision-making processes by retaining only relevant criteria in the principal components; thereby clearly outlining which criterion belongs to which factor. Since sparsity is desirable as it often leads to more interpretable results, reduced computation time, and improved generalization [44], sparse PCA-based methods are widely used for feature selection and clustering [45–48]. We will use sparse PCA as it can support the selection of the criteria for each principal components (PCs) [24].

Therefore, the proposed P-sPCA method facilitates the interpretation and visualization of loading factors and produce similar results to the proven GAIA method.

Method C) **P-SRD**: Qualifying the consistency of the criteria by Sum of Ranking Differences. The P-SRD method improves Promethee by evaluating the similarities between the criteria. The original SRD method calculates the sum of the ranking differences of the criteria to a golden standard ranking [49]. When this golden standard ranking is unknown, it is reasonable to take the average of the criteria provided that all criteria are measured on the same (or similar) scale. In this work, we will show that SRD can be considered as a special case of the Promethee method and the visualization of the ranking differences enables the structure of multicriteria decision-making problems to be analyzed.

## The three improvements of the Promethee-GAIA methodology

This section discusses the methodological approach of multicriteria decision-making solution techniques in the case of numerous criteria. The concepts of improving the Promethee-GAIA methodology (see Fig 1) are discussed more precisely in the following subsections.

**Details of the Promethee-GAIA methodology.** Promethee is addressed to tackle multi-criteria problem of ranking the $A$ set of possible alternatives $a_1, a_2, \ldots, a_m$ based on the $G$ set of evaluation criteria, $G$: $\{g_1(.), g_2(.), \ldots, g_n(.)\}$, where the criteria are weighted by the $\omega$: $(\omega_1, \omega_2, \ldots, \omega_k, \ldots, \omega_n)$ set of weights representing the importance of the criteria, $\sum_{k=1}^{n} \omega_k = 1$.

Fig 2 defines the methodological steps of the Promtehee method.

The preference degree refers to how an alternative is preferred against another. Preferences allocated to alternatives are based on the differences between the evaluations of the alternatives on a particular criterion:

$$d_k(a_i, a_j) = g_k(a_i) - g_k(a_j).\tag{1}$$

The preferences are calculated based on the mapping of the differences between 0 and 1:

$$P_k(a_i, a_j) = F_k[d_k(a_i, a_j)], \quad \forall a_i, a_j \in A \tag{2}$$

The method allows the flexible design of the preference function $F_k(d_k)$ for each criterion. Originally, six types of particular preference functions were proposed, from which the two types are shown in Fig 3.

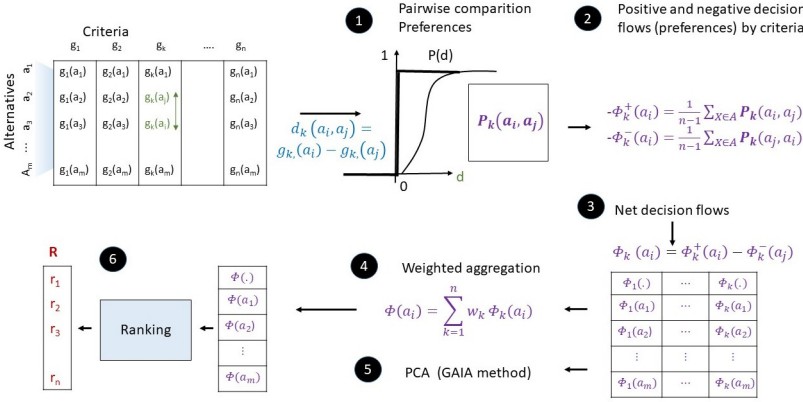

**Fig 2. Methodological steps of the Promethee method.**

| Generalised criterion | Definition | Parameters to fix |
|---|---|---|
| 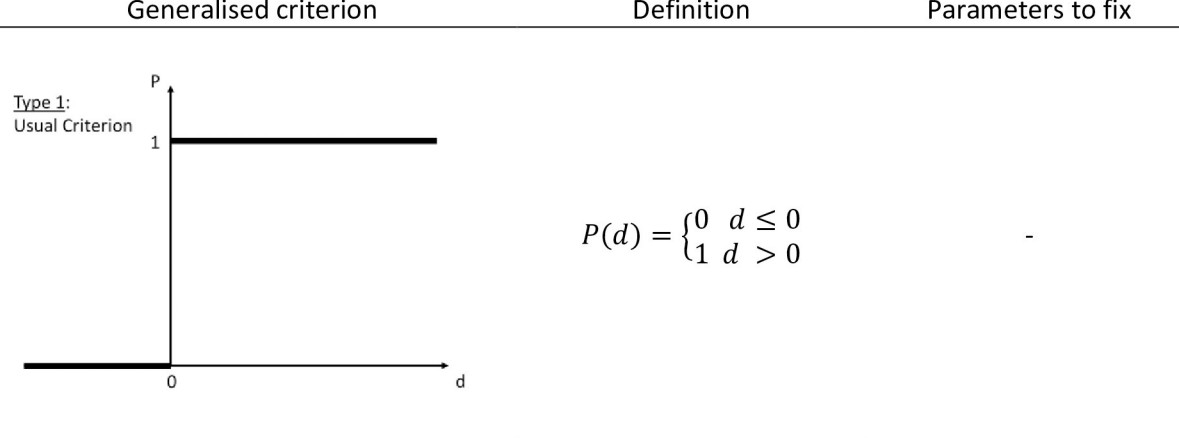 | $$P(d) = \begin{cases} 0 & d \leq 0 \\ 1 & d > 0 \end{cases}$$ | - |
| 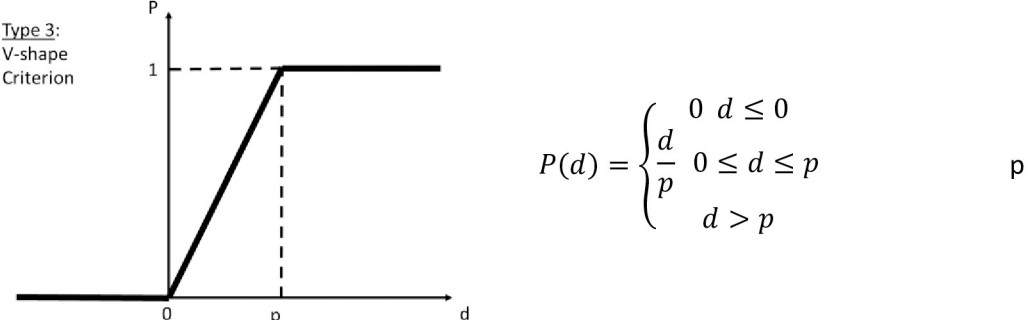 | $$P(d) = \begin{cases} 0 & d \leq 0 \\ \dfrac{d}{p} & 0 \leq d \leq p \\ & d > p \end{cases}$$ | $p$ |

**Fig 3. Types of generalised criteria (P(d)—Preference function).**

The $\phi_k^+(a_i) = \frac{1}{m-1}\sum_{j=1}^{m} P_k(a_i, a_j)$ positive outranking flows of each alternative shows how much an alternative is preferred over all the others. The $\phi_k^-(a_i) = \frac{1}{m-1}\sum_{j=1}^{m} P_k(a_j, a_i)$ negative outranking flow shows how all the other alternatives are preferred over one particular alternative. The net outranking flow shows the balance between the positive and negative outranking flows. The $\phi_k(a_i) = \phi_k^+(a_i) - \phi_k^-(a_i)$ is the single criterion net flow which expresses how an alternative $a_i$ outranks or is outranked by all other alternatives on criterion $g_k(.)$

$$\phi_k(a_i) = \phi_k^+(a_i) - \phi_k^-(a_i) = \frac{1}{m-1}\sum_{j=1}^{m}[P_k(a_i, a_j) - P_k(a_j, a_i)]. \tag{3}$$

The higher the net flow, the better the alternative. The aggregated decision can be obtained as the weighted sum of single criterion net flows:

$$\phi(a_i) = \sum_{k=1}^{n}\phi_k(a_i)\omega_k \tag{4}$$

The $\omega_k$ weights have a significant influence on the final ranking. By tuning the weights, the decision-maker can avoid redundant criteria causing an unbalanced shift in the multicriteria

decision in an undesired direction. Therefore, a careful analysis should be performed to highlight the hidden structure of the criteria.

The GAIA method provides a graphical representation of the **M** matrix of the unicriterion net flows:

$$\mathbf{M} = \begin{bmatrix} \phi_1(a_1) & \ldots & \phi_k(a_1) & \ldots & \phi_n(a_1) \\ \vdots & \vdots & \vdots & \vdots & \vdots \\ \phi_1(a_i) & \ldots & \phi_k(a_i) & \ldots & \phi_n(a_i) \\ \vdots & & \vdots & \vdots & \vdots \\ \phi_1(a_m) & \ldots & \phi_k(a_m) & \ldots & \phi_n(a_m) \end{bmatrix} \quad (5)$$

The purpose of the GAIA approach is to highlight the structure of the decision problem by applying the Principal Component Analysis (PCA) of the **M** matrix [50].

An illustrative GAIA plot (PCA biplot) is depicted in Fig 4. Points on the GAIA plane represent alternatives. The decision-maker gains information about how good the alternatives are on a particular criterion are and how similar such alternatives are. The closer the alternatives are to each other, the more similar they are, moreover, suitable alternatives to a particular criterion are located in the direction of the criterion axis [20]. On the biplot, the loading vectors of the criteria are also shown. The length of the vectors refer to how discriminant the criterion is (the longer the axis, the higher the discriminating power). If the decision power is strong, alternatives should be selected to its direction. The angles between the vectors tell us how the criteria correlate with one another. When two vectors are close, forming a small angle, the two

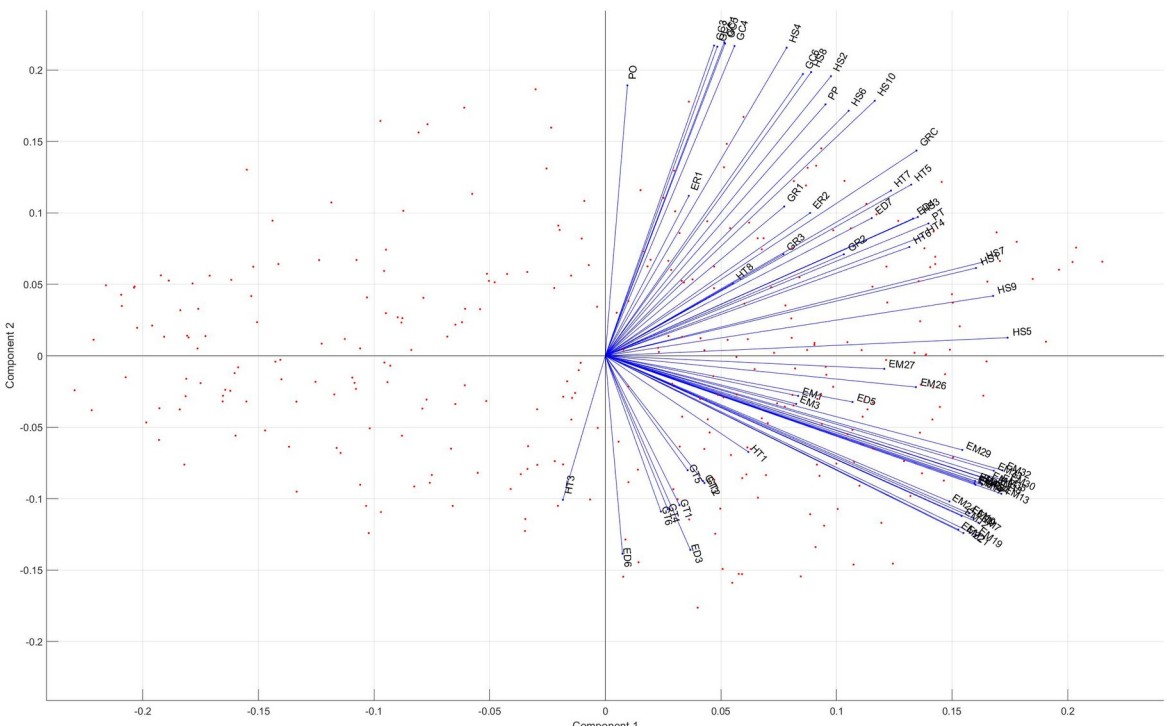

**Fig 4. The Promethee-GAIA plane of the I4.0+ criteria—Red dots represents the alternatives and blue vectors denote the loading vectors of the criteria.**

criteria tend to correlate positively. If they intercept at 90 degrees to each other, they are unlikely to correlate. When they diverge away from each other and form a large angle (close to 180 degrees), they are negatively correlated. In that case, the criteria are strongly conflicting.

Based on the highlighted information, the decision-maker can become aware of the relationships and similarities between criteria by fine-tuning of their weights.

One drawback of the GAIA visual decision aid is that it relies on PCA, which allows all original criteria to form the principal components and often makes the interpretation difficult. Furthermore, the linear combination of criteria does not separate criteria clearly regarding the creation of factors. Methods for better and more transparent selection of criteria will be introduced to enhance the factor structure, thereby improving interpretability.

**Method A) Principal Factor Analysis with rotations and communality analysis (P-PFA).** Common factor analysis, also referred to as Principal Factor Analysis (PFA) or principal axis factoring, seeks the fewest factors, which can account for the common variance (correlation) between a set of criteria. The aim of PFA is twofold: (1) to group criteria into latent factors, while retained account for a maximum amount of variance between observed criteria; as well as to ignore (2a) redundant (in this case, multicollinear) and (2b) irrelevant criteria [41]. PCA can be explained by the equation: $\mathbf{Z} = \mathbf{FP}$, where $\mathbf{Z}$ denotes the $n$ by $m$ standardized original data matrix of $\mathbf{M}$, $\mathbf{F}$ represents the standardized factor score matrix and $\mathbf{P}$ stands for the factor $\times$ criterion weight matrix [41]. Columns of $\mathbf{P}$ are multiplied by the square root of corresponding eigenvalues, that is, eigenvectors scaled up by the variances. To put it simply, it is assumed that $\mathbf{Z}$ represents the variance between criteria, $\mathbf{F}$ denotes the variance of common factors (or a common variance of a $\mathbf{Z}$ criterion with other analyzed criteria) and $\mathbf{P}$ stands for coefficients showing how $\mathbf{F}$ and $\mathbf{Z}$ are related [41]. It should be noted that common variance (or covariance) can be considered as types of correlation for simpler understanding. The PFA equation is: $\mathbf{Z} = \mathbf{FP} + \mathbf{U}$. The difference between the two equations is the last component (*i.e.*, $\mathbf{U}$). $u_j \in \mathbf{U}$ represents the unique variance of a criterion $j$. PCA assumes that the communality ($h_i^2 = \sum_{j=1}^{k} p_{ij}^2, p_{ij} = [\mathbf{P}]_{ij}$), *i.e.*, common variance, becomes maximized and no unique variance of each criterion is present, whereas PFA assumes a substantial amount of unique variance ($u_i = 1 - h_i^2$). PCA projects original criteria into a smaller number of components, that is, model or dimensionality reduction. PFA identifies a factor model (factor structure) that would best reproduce the observed correlation between criteria, thereby aiming to explain the correlation [41]. In summary, although both PCA and PFA specify the group of criteria, the difference between a component of a PCA and factor of a PFA can be stated as the following: criteria specify components, while factors specify relevant criteria [41, 51]. Since PFA specifies criteria, the method usually involves a the selection of critera, where multicollinear and mismatched criteria have to be ignored.

The interpretation of a factor depends on which group of criteria are correlated to the factor, dominantly. In other words, the interpretation of a factor determines which criteria belong to the factor. Therefore, an important requirement is that it can be clearly decided to which factor a criterion belongs, otherwise the factors are difficult to interpret [52].

A criterion which has a low communality value does not fit any factor, in other words, it is *irrelevant to the model* so should be dropped. Both PCA and PFA assumes the original criteria are correlated to each other. To ensure that the correlations among criteria are sufficiently strong, the Kaiser-Meyer-Olkin (KMO) [42] method is used to test the relationships among criteria. It measures how suitable data is for factor analysis based on Measure of Sampling Adequacy (MSA) [42]. Sampling adequacy predicts, if data are likely to factor well, based on correlations and partial correlations. The overall MSA, and the MSA for the criterion $j$ can be

computed, respectively, as follows:

$$MSA = \frac{\sum \sum_{i\neq j} r_{ij}^2}{\sum \sum_{i\neq j} r_{ij}^2 + \sum \sum_{i\neq j} q_{ij}^2}, MSA_j = \frac{\sum_{i\neq j} r_{ij}^2}{\sum_{i\neq j} r_{ij}^2 + \sum_{i\neq j} q_{ij}^2},$$ (6)

where $r_{ij}$ denotes the correlation coefficient and $q_{ij}$ stands for the partial correlation coefficient between criterion $i$ and criterion $j$. If criteria are multicollinear, the partial correlation coefficient will be high, thereby reducing the value of the MSA.

On the one hand, since the value of $MSA_j$ depends on how many criteria are implemented, it is used to assess, which criteria should to be dropped from the model because they are too multicollinear (redundant). On the other hand, given that the value of MSA does not depends on how many factors are used in the model, it should be calculated for all criteria before the PFA.

Both PCA and PFA are model reduction techniques, where only the first $k < n$ components / factors are retained. Therefore, the explained variance ratio ($v_k$), which is the proportion of the amount of variance explained by each of the first $k$ factors per the total amount of variance, should be greater than a threshold. In the practice, usually $v_k > 0.5$.

Before rotating factors, the variance explained is the highest for the first factor and decreases for the remaining factors. Nevertheless, without rotating factors, most of the criteria belong to the first factor, which renders interpretation difficult. Therefore, a varimax rotation method is used to balance the explained variances between the factors [43]. It is so called varimax method because it maximizes the sum of the variances of the squared loadings (squared correlations between criteria and factors). Preserving orthogonality requires a rotation to leave the sub-space invariant.

The interpretation of factors is also difficult given that a criterion may belong to more than one factor. $c_{min}$ denotes the minimum difference in correlation between two criteria. Without restricting of completeness, for criterion $j$, suppose that $|p_{j1}| \geq |p_{j2}| \geq .. \geq |p_{jk}|$ is satisfied. Criterion $j$ is not a common criterion, if either $|p_{j1}| > |p_{j2}| + c_{min}$ or $|p_{j1}| > 2|p_{j2}|$, otherwise criterion $j$ is a common criterion. In an iteration, the common criterion which has lowest communality value is ignored [52]. This iteration finishes once there are no more common criteria. In order to improve the factor structure for the purpose of enhancing interpretation, PFA applies the following criterion selection techniques. These criterion selection techniques run iteratively.

1. *Ignoring multicollinear criteria.* Before PFA, criterion $j$ should be ignored, if $MSA_j$ is below a predefined threshold. Therefore, in all the steps, the criterion with the lowest $MSA_j$ value is ignored, until every $MSA_j$ and the overall $MSA$ are greater than the threshold ($MSA_{min}$).

2. *Ignoring irrelevant criteria.* The criteria, which predominantly do not fit any factors must be ignored. In an iteration, the criterion with the lowest communality value is dropped until every communality value is greater than a threshold ($h_{min}$).

3. *Ignoring common criteria.* If every criterion is relevant, then just common criteria which has the lowest community values should be ignored. The iteration should be continued until a common criterion for a given $c_{min}$ threshold is identified.

P-PFA also uses varimax rotation in order to decrease the number of common criteria. The varimax method is an orthogonal rotation method that tends to produce factor loadings that are either very high or very low, making it easier to match each criterion to a single factor. With varimax rotation and criterion selections, the PFA specifies (1) not multicollinear (2) relevant and (3) noncommon (unique) criteria.

**Method B) integration of sparse PCA into the Promethee method (P-sPCA).** Sparse Principal Component Analysis [24] is based on the idea of combining the advantages of PCA with an elastic net regression formulation to create sparse loading vectors. Elastic net regression is used to regress the loading vectors of the original PCA. The target of the analysis determines how the tuning parameters per PC should be set. A distinctioncan be made between three simple targets:

- resulting as few PCs as possible: the correlation between non-zero loadings should be maximized to ensure that the explained variances are as high as possible;

- keep PCs uncorrelated as possible: the number of non-zero loadings should be decreased (loadings with smaller values than a threshold considered to be zero);

- improve the interpretability of PCs: the number of non-zero loadings should be decreased, so less alternatives are loaded into the same PC, thereby making the explanation easier.

This section highlights the utilization potential of the third target, that is 'making PCs more interpretable', by implementing the sparse PCA method into the Promethee-GAIA methodology. The Promethee-GAIA methodoly is a visual decision aid based on PCA and projects information in a $k$-dimensional (sub)space on a hyperplane. However, the visualization of a high-dimensional dataset can be critical as the PCs are the linear combination of all criteria, each of which form the principal components. In this case, the sparse PCA method simplifies the interpretation by selecting fewer criteria to form the PCs.

However, some implications need to be considered when moving from PCA to sparse PCA. Scores and loadings in sparse PCA may not be orthogonal; therefore, the traditional way of computing scores, residuals and explained variance in terms of PCA can be misleading resulting in unexpected properties and an incorrect interpretation as well as affect the visualization [53].

The proposed approach of integrating sparse PCA into the Promthee-GAIA can methodology promote statistical analysis and create composite indices by supporting groupings and feature selections. Studies underline the utilization efficiency of the sparse PCA method with regard to summarizing and organizing extensive text data [48], clustering and problems concerning the feature selection of gene expression data analysis [45], as well as optimization with grouping constraints in process monitoring sensor networks and social networks [54].

The sPCA methodology is formulated *via* elastic net regression, where the basic idea is to combine the advantages of PCA with the formulation of an elastic net regression to create sparse loading vectors which only depend on the most critical physical criteria (measurements). Elastic net regression is used to regress the loading vectors of the original PCA [24] as follows:

$$\mathbf{b}_{sparse} = \underset{\mathbf{b}}{\operatorname{argmin}} \|\mathbf{P}_i - \mathbf{Mb}\|^2 + \lambda \|\mathbf{b}\|^2 + \lambda_1 \|\mathbf{b}\|_1 \qquad (7)$$

where $\mathbf{P}_i$ denotes the $i^{th}$ PC of the original PCA and $\mathbf{b}$ stands for the sparse loading vector. Sparseness is achieved through the parameters $\lambda$ and $\lambda_1$, which penalize all non-zero loadings in $\mathbf{b}$. For $\lambda, \lambda_1 = 0$ the regression problem simplifies and the obtained loadings $\mathbf{b}$ reduce to PCA loadings. The $L_1$ norm for $\lambda_1$ and $L_2$ norm for $\lambda$ are used in this formulation. The $L_2$ norm is necessary when the number of predictors $p$ exceeds the number of observations ($p \gg n$).

Sparse PCA algorithms optimize the non-zero loadings (NZL) to capture the variance with as few PCs as possible. The second goal of sPCA is making PCs interpretable by decreasing the number of NZL making the explanation of the model easier. A third goal is to keep each PC as

uncorrelated as possible which can be achieved also by decreasing the number of NZLs. The Index of Sparseness is is formulated according to these goals:

$$IS = \frac{V_a \ V_s}{V_o^2} \frac{\#_0}{n \times p}$$

(8)

with $V_a$, $V_s$ and $V_o$ denoting the adjusted, unadjusted and ordinary total variances, respectively, $\#_0$ stands for the number of zero loadings and $n$ and $p$ represent the number of criteria and alternatives, respectively. It is clear that $\left(\frac{\#_0}{n \times p}\right)$ is maximized by the increasing number of zero loadings, while $\frac{V_a}{V_o}$ and $\frac{V_s}{V_o}$ denote the increasing fraction of explained variance. This index will be used as a metric to determine the desirable number of NZLs per PC.

**Method C) analysis of the criteria based on the Sum of Ranking Differences (P-SRD) method.** The Sum of Ranking Differences method, abbreviated as SRD, supports multicriteria decision-making as it can measure the correlation between (consistency of) criteria. The P-SRD method utilizes this aspect to explore more information about the similarities between and groupings of criteria.

SRD is suitable for the fair comparison of methods and models. It is a city block (Manhattan) distance between the sum ranked item differences, if a gold standard (benchmark) is fixed for the comparison. Easily understandable explanation and practical examples have already been published [49, 55–57].

The methodological steps of P-SRD are shown in Fig 5:

The input matrix of SRD is the original data matrix of Promethee (evaluation table), where the alternatives $A$: $\{a_1, a_2, \ldots, a_m\}$ located in rows and criteria $G$: $\{g_1, g_2, \ldots g_n\}$ are arranged in columns representing the criteria. In this regard, $g_k(a_i)$ represents the value of $a_i$ alternative regarding the $g_k$ criterion. The original data matrix should be standardized or re-scaled before

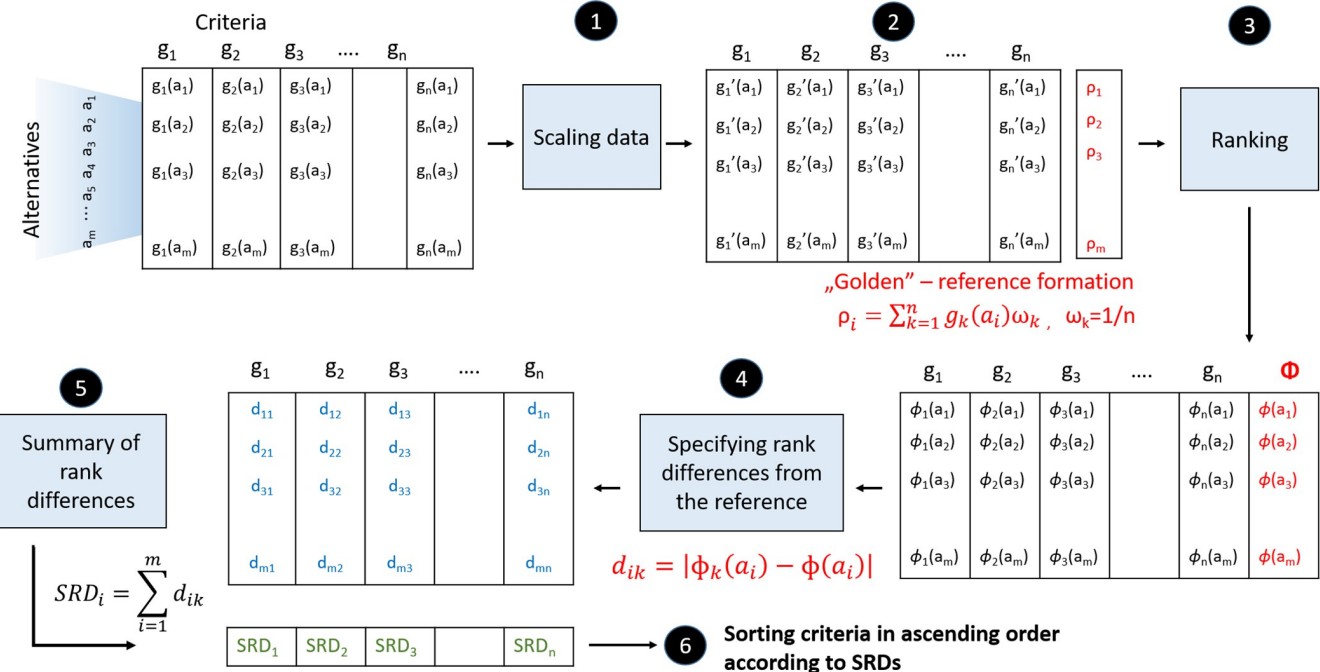

**Fig 5. Methodological steps of the P-SRD method.**

the analysis to ensure that criteria are measured on the same scale. The last column of the matrix contains the benchmark values, that is, the references, which are the basis of the comparison. In terms of ranking, the most frequently used benchmark is the average [58]. In the case of the average, SRD measures the difference from the centre; it is a non parametric measure of similarity (or its reverse the similarity) [56]. Generally, the average can be accepted as a benchmark in the absence of a known standard as the errors cancel each other out. "The maximum likelihood principle will ensure that the most probable ranking will be provided by the average. If a reference ordering is known, then not the average, but the given benchmark sequence should be used for comparison and calculation of absolute ranking differences" [56].

If the golden standard $\rho_i$ is the average, the row average is calculated as follows (the weight assigned to the criteria is $\omega = \frac{1}{n}$):

$$\rho_i = \sum_{k=1}^{n} g_k(a_i)\omega_k, \quad \omega_k = \frac{1}{n} \tag{9}$$

The SRD values are then calculated by taking the absolute values of the differences between the reference ranking (golden standard) and the individual ones, which can be described as $d_{ik} = |\phi_k(a_i) - \phi(a_i)|$. The absolute differences are then totalled for each criterion, $\Sigma d_{ik}$. Subsequently, criteria are sorted in ascending order according to the SRD values. The visualisation occurs in a one-dimensional space, where the normal approximation values are read on the right abscissa, while the SRD% values are shown on the left abscissa and on the ordinate [59]. The closer the SRD value is to the golden standard, the better the criterion. Closely proximate SRD values represent close similarities between the criteria.

The connection between the SRD and Promethee methods provides ranking organisation and forms criteria. In a special case the SRD method takes the average formed from the criteria as the gold standard. Promethee is based on the average of the preferences formed from the criteria.

Therefore, concerning Promethee, a targeted ranking can be created for the SRD of each criterion. It is believed that the distribution of Manhattan distances used in the SRD method and its analysis are suitable to qualify the consistency of multicriteria decision tasks and models. Both methods are widely used in statistical analysis and the evaluation of the composite criterion to support decision-making.

Empirical evidences suggests that SRD is an easily perceivable multicriteria decision-making tool. Recent examinations have clearly and unambiguously shown that the sum of ranking differences (SRD) yields a multicriteria optimisation. In references [60, 61], SRD was used as an MCDM technique. Lourenço and Lebensztajn have illustrated in two practical examples that SRD provides a consensus of eight MCDM methods without using any subjective weights [62]. Other MCDM methods regard various parts of the Pareto front as optimal. A recent paper [63] compares resampling methods to avoid erroneous bootstrapping and suggests using the analysis of variance (ANOVA) of SRD values to obtain a better overall picture for comparison [63]. All details of calculations and validation tests can be found in Refs. [49, 55, 56].

## Results and discussion

### Application study

This section compares the applicability of the proposed methods through a case study in which the regional Industry 4.0 (I4.0+) readiness of economic regions is measured.

Industry 4.0-related developments require the utilization of MCDM methods and their targeted improvement. The importance of decision-support methods in terms of measuring Industry 4.0 readiness of countries [64, 65], regions [66–68], cities [69], and enterprises [70–72] has been emphasized by previous literature. Industry 4.0 readiness models and decision-support methods enable decision-makers to formulate strategic plans based on I4.0-specific metrics. Strategic plans can be made for e.g. supplier selection or maintenance strategies [73], quality management strategies [74], investments and (regional) development strategies [75]. Defining and measuring regional development status through an open indicator system and utilizing decision-support methods can provide comprehensively available and applicable information [75]. Thereby, the opportunity is given for each region to evaluate and compare their status and define innovation and development strategies to adapt to Industry 4.0 successfully [76].

The analyzed I4.0+ data is a subset of open data that analyzes NUTS 2-classified regional Industry 4.0 readiness [75].

The I4.0+ dataset relies on the following open data sources: ETER, Erasmus, USPTO, MA-Graph, GRID, Eurostat and The GDELT Project. The initially constructed dataset includes 414 alternatives (regions) and 101 criteria, categorized into five dimensions: higher education and lifelong learning, innovation, investment, the labour market, and technology. A more detailed description of the data is available in Ref. [77] and all the original data, definitions of cirteria and rankings can be accessed from the following data repository link: https://data.mendeley.com/datasets/23gwn43ygp/1. It must be noted that 34.11% of the original dataset is missing. Therefore, the data was carefully selected. Alternatives with previous (invalid) NUTS 2 codes and ones with at least 35% of the values missing were eliminated from the dataset. Criteria with at least 40% of the values missing were also dropped. Although most of the eliminated criteria are sub-criteria, two main criteria were eliminated due to the lack of data, namely 'Intramural R&D expenditure (GERD) by sectors of performance and NUTS 2 regions' and the 'Total R&D personnel and researchers by sectors of performance, sex and NUTS 2 regions'. In this way, financial initiatives and investments in I4.0 cannot be measured by the model.

The analyzed dataset includes 69 criteria and 314 alternatives. A short description of the selected variables is included in S1 Appendix.

During the analysis of the data, firstly the Promethee-based matrix of criterion net flows is determined, then the P-PFA, P-sPCA and P-SRD methods are applied to the flow matrix. Similarities between criteria can be explored more precisely, identifying which criteria belong to which factor and eliminating redundant criteria. Therefore, more characteristic relationships between criteria and more interpretable results are produced.

**Application of method A) principal factoring with rotation and communality analysis (P-PFA).** Principal factoring with rotation and communality analysis was applied to the Promethee-based matrix of criterion net flows. The method eliminated 34 multicollinear criteria out of the 69 variables.

Fig 6 shows the criteria selected by the P-PFA method including the abbreviations of the criteria and the colour-coded values of its two factor loadings. The greener the cell, the higher the value.

It is worth noting that the criteria are separated into two factors based on the values of factor loadings. These two factors are the following:

1. Employment rates and job opportunities.
   The first group involves the 'Employment rates of young people not in education and training by sex, educational attainment level, years since completion of the highest level of

| Name of the criterion | | Criterion | Loading 1 | Loading 2 |
|---|---|---|---|---|
| Employment rates of young people not in education and training by sex, educational attainment level, years since completion of highest level of education and NUTS 2 regions | ED0-4 Less than primary, primary, secondary and post-secondary non-tertiary education (levels 0-4); Duration: Total; Age 15-34; PC (%) | 'EM7' | 0,96 | 0,01 |
| Employment rates of young people not in education and training by sex, educational attainment level, years since completion of highest level of education and NUTS 2 regions | ED0-4 Less than primary, primary, secondary and post-secondary non-tertiary education (levels 0-4); Y_GT3: over 3 years; Age 15-34; PC (%) | 'EM9' | 0,95 | 0,01 |
| Employment rates of young people not in education and training by sex, educational attainment level, years since completion of highest level of education and NUTS 2 regions | ED0-4 Less than primary, primary, secondary and post-secondary non-tertiary education (levels 0-4); Y_GT3: over 5 years; Age 15-34; PC (%) | 'EM10' | 0,93 | 0,02 |
| Employment rates of young people not in education and training by sex, educational attainment level, years since completion of highest level of education and NUTS 2 regions | ED0-4 Less than primary, primary, secondary and post-secondary non-tertiary education (levels 0-4); Y_LE3: 5 years or less; Age 15-34; PC (%) | 'EM12' | 0,91 | 0,04 |
| Employment rates of young people not in education and training by sex, educational attainment level, years since completion of highest level of education and NUTS 2 regions | ED3-8 Upper secondary, post-secondary non-tertiary and tertiary education (levels 3-8); Duration: Total; Age 15-34; PC (%) | 'EM13' | 0,97 | 0,07 |
| Employment rates of young people not in education and training by sex, educational attainment level, years since completion of highest level of education and NUTS 2 regions | ED3-8 Upper secondary, post-secondary non-tertiary and tertiary education (levels 3-8); Y1-3:from 1 to 3 years; Age 15-34; PC (%) | 'EM14' | 0,89 | 0,10 |
| Employment rates of young people not in education and training by sex, educational attainment level, years since completion of highest level of education and NUTS 2 regions | ED3-8 Upper secondary, post-secondary non-tertiary and tertiary education (levels 3-8); Y_GT3: over 3 years; Age 15-34; PC (%) | 'EM15' | 0,93 | 0,06 |
| Employment rates of young people not in education and training by sex, educational attainment level, years since completion of highest level of education and NUTS 2 regions | ED3-8 Upper secondary, post-secondary non-tertiary and tertiary education (levels 3-8); Y_GT3: over 5 years; Age 15-34; PC (%) | 'EM16' | 0,91 | 0,04 |
| Employment rates of young people not in education and training by sex, educational attainment level, years since completion of highest level of education and NUTS 2 regions | ED3-8 Upper secondary, post-secondary non-tertiary and tertiary education (levels 3-8); Y_LE3: 3 years or less; Age 15-34; PC (%) | 'EM17' | 0,89 | 0,10 |
| Employment rates of young people not in education and training by sex, educational attainment level, years since completion of highest level of education and NUTS 2 regions | ED3-8 Upper secondary, post-secondary non-tertiary and tertiary education (levels 3-8); Y_LE3: 5 years or less; Age 15-34; PC (%) | 'EM18' | 0,94 | 0,11 |
| Employment rates of young people not in education and training by sex, educational attainment level, years since completion of highest level of education and NUTS 2 regions | ED3_4 Upper secondary and post-secondary non-tertiary education (levels 3 and 4); Duration: Total; Age 15-34; PC (%) | 'EM19' | 0,97 | -0,01 |
| Employment rates of young people not in education and training by sex, educational attainment level, years since completion of highest level of education and NUTS 2 regions | ED3_4 Upper secondary and post-secondary non-tertiary education (levels 3 and 4); Y_GT3: over 3 years; Age 15-34; PC (%) | 'EM21' | 0,94 | -0,03 |
| Employment rates of young people not in education and training by sex, educational attainment level, years since completion of highest level of education and NUTS 2 regions | ED3_4 Upper secondary and post-secondary non-tertiary education (levels 3 and 4); Y_GT3: over 5 years; Age 15-34; PC (%) | 'EM22' | 0,93 | -0,04 |
| Employment rates of young people not in education and training by sex, educational attainment level, years since completion of highest level of education and NUTS 2 regions | ED3_4 Upper secondary and post-secondary non-tertiary education (levels 3 and 4); Y_LE3: 5 years or less; Age 15-34; PC (%) | 'EM24' | 0,86 | 0,08 |
| Employment rates of young people not in education and training by sex, educational attainment level, years since completion of highest level of education and NUTS 2 regions | ED5-8 Tertiary education (levels 5-8); Y_LE3: 5 years or less; Age 15-34; PC (%) | 'EM29' | 0,82 | 0,17 |
| Employment rates of young people not in education and training by sex, educational attainment level, years since completion of highest level of education and NUTS 2 regions | ED Total; Duration: Total; Age 15-34; PC (%) | 'EM30' | 0,97 | 0,09 |
| Employment rates of young people not in education and training by sex, educational attainment level, years since completion of highest level of education and NUTS 2 regions | ED Total; Y1-3 : from 1 to 3 years; Age 15-34; PC (%) | 'EM31' | 0,90 | 0,10 |
| Employment rates of young people not in education and training by sex, educational attainment level, years since completion of highest level of education and NUTS 2 regions | ED Total; Y_GT3: over 3 years; Age 15-34; PC (%) | 'EM32' | 0,95 | 0,10 |
| Employment rates of young people not in education and training by sex, educational attainment level, years since completion of highest level of education and NUTS 2 regions | ED Total; Y_GT3: over 5 years; Age 15-34; PC (%) | 'EM33' | 0,93 | 0,09 |
| Employment rates of young people not in education and training by sex, educational attainment level, years since completion of highest level of education and NUTS 2 regions | ED Total; Y_LE3: 3 years or less;Age 15-34; PC (%) | 'EM34' | 0,91 | 0,11 |
| Employment rates of young people not in education and training by sex, educational attainment level, years since completion of highest level of education and NUTS 2 regions | ED Total; Y_LE3: 5 years or less; Age 15-34; PC (%) | 'EM35' | 0,94 | 0,12 |
| Human Resources in Science and Technology by category and NUTS 2 regions | HRSTO- Persons employed in science and technology; PC_POP - Percentage of total population | 'HS5' | 0,77 | 0,33 |
| Human Resources in Science and Technology by category and NUTS 2 regions | SE- Scientists and engineers; PC_POP - Percentage of total population | 'HS9' | 0,70 | 0,33 |
| Human Resources in Science and Technology by category and NUTS 2 regions | HRSTC- Persons with tertiary education (ISCED) and employed in science and technology; THS - Thousand | 'HS2' | 0,15 | 0,98 |
| Human Resources in Science and Technology by category and NUTS 2 regions | HRSTE- Persons with tertiary education (ISCED); THS-Thousand | 'HS4' | 0,04 | 0,99 |
| Human Resources in Science and Technology by category and NUTS 2 regions | HRSTO- Persons employed in science and technology; THS - Thousand | 'HS6' | 0,23 | 0,95 |
| Human Resources in Science and Technology by category and NUTS 2 regions | HRST- Persons with tertiary education (ISCED) and/or employed in science and technology; THS - Thousand | 'HS8' | 0,11 | 0,99 |
| Human Resources in Science and Technology by category and NUTS 2 regions | SE- Scientists and engineers; THS - Thousand | 'HS10' | 0,27 | 0,94 |
| Number of research-related institutions | | 'GRC' | 0,39 | 0,73 |
| Number of news in 'competitive industries' | | 'GC1' | -0,03 | 0,62 |
| Number of news in 'education skills development and labour market' | | 'GC2' | -0,04 | 0,59 |
| Number of news in 'employability skills and jobs' | | 'GC3' | -0,05 | 0,59 |
| Number of news in 'industry policy' | | 'GC4' | -0,01 | 0,62 |
| Number of news in 'jobs' | | 'GC5' | -0,03 | 0,63 |
| Number of news in 'manufacturing' | | 'GC6' | 0,14 | 0,66 |

**Fig 6. Criteria selected by the P-PFA method—The loading values are colour coded, the greener, the higher the value.**

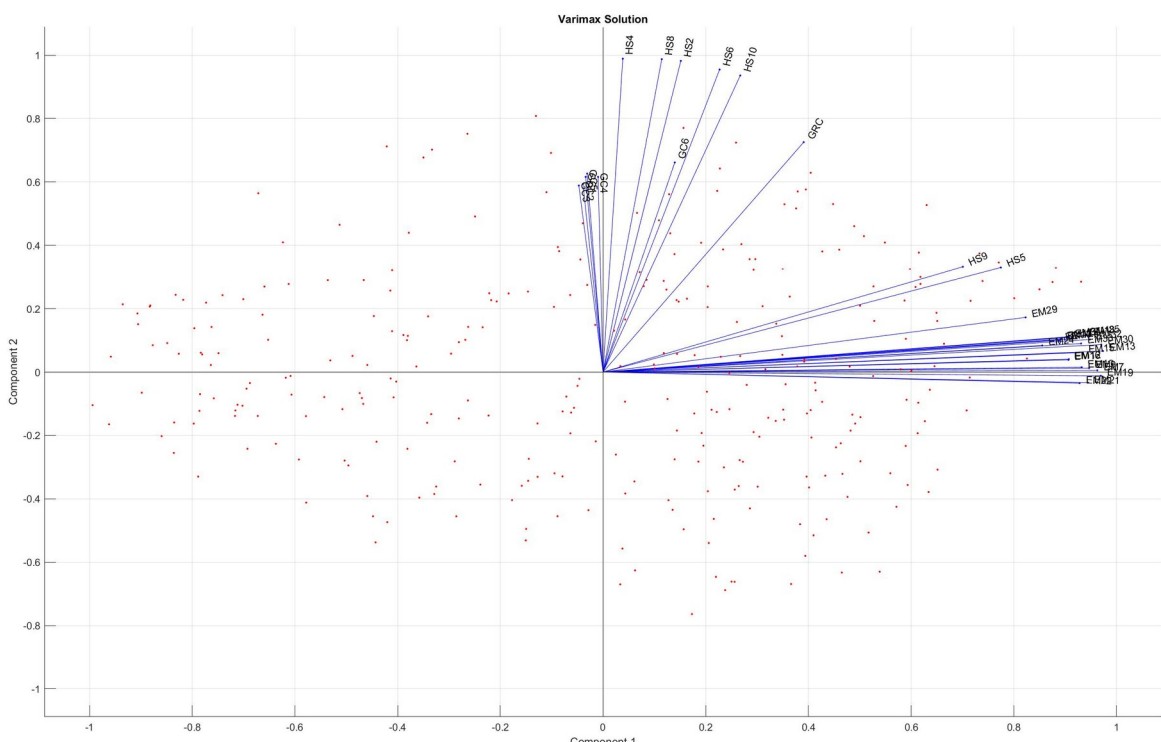

**Fig 7. Interpretation of criteria selected by the PFA method indicating two major criterion groupings: 1) employment rates of young people measured according to the years since their highest level of education was completed and 2) human resources in the fields of science and technology coupled with the number of research institutions in and news appearings concerning I4.0-related fields.**

education and NUTS 2 regions' criterion with its sub-criteria (EM7-EM35). This criterion measures the number of people (aged between 15 and 34) employed after finishing their education and categorizes them according to the number of years passed since completing their highest level of education.

We can assume that where the employment rates of young people are higher (after finishing their education less than 2–5 years), job opportunities exist and economic value is generated in a region, fostering adaptation capability [75].

2. Academia sector and its employees in R&D fields coupled with the number of I4.0-related news appearings.

The second group of criteria includes the criteria and sub-criteria of 'Human Resources in Science and Technology by category and NUTS 2 regions' (HS2-HS10), the 'Number of research-related institution' (GRC), the 'Number of news appearings in 'competitive industry' (GC1), 'education skills development and labour market' (GC2), 'employability skills and jobs' (GC3), 'industry policy' (GC4), 'jobs' (GC5) and 'manufacturing' (GC6). This factor can reflect the regional capability of innovation adaptation.

Fig 7 represents the biplot of the model using varimax rotation, which visually underlines the clear separation of the two groups of criteria. Significant differences can be found by comparing Figs 4 to 7. The criteria groups and their separation is better identified in P-PFA (1) employment rates and job opportunities; 2) academia sector and its employees in R D fields coupled with the number of I4.0-related news appearings). According to the original GAIA-

| Name of the criterion | | PC1 | Loadings |
|---|---|---|---|
| Population aged 30-34 by educational attainment level, sex and NUTS 2 regions (%) | ED3-8 Upper secondary, post-secondary non-tertiary and tertiary education (levels 3-8); sex:Total | 'ED5' | 0,307292 |
| Population aged 30-34 by educational attainment level, sex and NUTS 2 regions (%) | ED3_4 Upper secondary and post-secondary non-tertiary education (levels 3 and 4); sex:Total | 'ED6' | 0,009334 |
| Population aged 30-34 by educational attainment level, sex and NUTS 2 regions (%) | ED5-8 Tertiary education (levels 5-8); sex:Total | 'ED7' | 0,349714 |
| Employment rates of young people not in education and training by sex, educational attainment level, years since completion of highest level of education and NUTS 2 regions | ED3_4 Upper secondary and post-secondary non-tertiary education (levels 3 and 4); Y_LE3: 5 years or less; Age 15-34; PC (%) | 'EM24' | 0,303887 |
| Employment rates of young people not in education and training by sex, educational attainment level, years since completion of highest level of education and NUTS 2 regions | ED5-8 Tertiary education (levels 5-8); Y_GT3: over 3 years; Age 15-34; PC (%) | 'EM26' | 0,288652 |
| Employment rates of young people not in education and training by sex, educational attainment level, years since completion of highest level of education and NUTS 2 regions | ED Total; Y1-3 : from 1 to 3 years; Age 15-34; PC (%) | 'EM31' | 0,345465 |
| Number of graduates in Engineering, Manufacturing and Construction (BSc, MSc, PhD) | | 'GR3' | 0,218016 |
| Human Resources in Science and Technology by category and NUTS 2 regions | HRSTC- Persons with tertiary education (ISCED) and employed in science and technology; PC_POP - Percentage of total population | 'HS1' | 0,416411 |
| Employment in technology and knowledge-intensive sectors by NUTS 2 regions and sex (from 2008 onwards, NACE Rev. 2) | KIS_HTC- Knowledge-intensive high-technology services; PC_EMP - Percentage of total employment; sex: Total | 'HT5' | 0,371578 |
| Number of research-related institutions | | 'GRC' | 0,359405 |

| Name of the criterion | | PC2 | Loadings |
|---|---|---|---|
| Population on 1 January by age, sex and NUTS 2 region | Age 1-99; sex: Total; Count | 'PO' | -0,30312 |
| Human Resources in Science and Technology by category and NUTS 2 regions | HRSTC- Persons with tertiary education (ISCED) and employed in science and technology; THS - Thousand | 'HS2' | -0,25755 |
| Human Resources in Science and Technology by category and NUTS 2 regions | HRSTE- Persons with tertiary education (ISCED); THS-Thousand | 'HS4' | -0,28791 |
| Human Resources in Science and Technology by category and NUTS 2 regions | HRST- Persons with tertiary education (ISCED) and/or employed in science and technology; THS - Thousand | 'HS8' | -0,27508 |
| Number of news in 'competitive industries' | | 'GC1' | -0,34372 |
| Number of news in 'education skills development and labour market' | | 'GC2' | -0,3449 |
| Number of news in 'employability skills and jobs' | | 'GC3' | -0,34488 |
| Number of news in 'industry policy' | | 'GC4' | -0,34054 |
| Number of news in 'jobs' | | 'GC5' | -0,34305 |
| Number of news in 'manufacturing' | | 'GC6' | -0,30585 |

**Fig 8. Criteria of PCs selected by the P-sPCA method and their corresponding loadings.**

plane the mosr relevant criteria groups cannot be clearly selected. Due to the varimax rotation in P-PFA, the location of factor loadings changed in Fig 7 compared to Fig 4, thereby alternatives which perform well on a particular criterion will differ as well as the complete ranking of alternatives.

**Application of method B) the integration of sparse PCA into the Promethee method (P-sPCA).** The sPCA method is applied to the Promethee-based criterion net flow matrix. The analysis is based on the spca_am toolbox.

Ten criteria per PC were selected out of 69. Fig 8 indicates these criteria which are included in the principal components (PC) as well as their loadings.

The two PCs are clearly separated and highlight diverse segments supporting the regional Industry 4.0 development. These two segments are the following:

PC 1: Academia sector and its employees coupled with the number of research institutions and the employment rate after finishing education.

The first PC includes the sub-criteria of the 'Population aged 30–34 by educational attainment level' (ED5–7) referring to the perspective on lifelong learning, the 'Employment rates of young people not in education and training' (EM24, EM26, EM31) that reveals information about job opportunities according to educational attainment levels, the 'Number of graduates in Engineering, Manufacturing and Construction (BSc, MSc, PhD)',

reflecting the future workforce in these fields, 'Human Resources in Science and Technology' (HS1) which defines the percentage of people in tertiary education and employed in the fields of science and technology, 'Employment in technology and knowledge-intensive sectors' (HT5) that defines the percentage of people employed in knowledge-intensive high-technology services, and the 'Number of research-related institutions' that provides a regional knowledgebase.

PC 2: People employed in the Science and Technology sectors coupled with the number of world news appearings concerning the field of I4.0.

The second PC involves the sub-criterion of 'Human Resources in Science and Technology' (HS2, HS4, HS8) which defines the thousands of people studying in tertiary education, employed in the fields of science and technology field as well as those studying in tertiary education whilst being employed in the fields of science and technology, the 'Number of news appearings concerning 'competitive industry' (GC1), 'education skills development and labour market' (GC2), 'employability skills and jobs' (GC3), 'industry policy' (GC4), 'jobs' (GC5) and 'manufacturing' (GC6).

**Application of method C) qualifying the consistency of the criteria by Sum of Ranking Differences (P-SRD).** The SRD method is applied on the Promethee-based criterion net flow matrix to qualify the consistency of the criteria.

Fig 9 shows the rank of criteria based on the P-SRD method. Criteria belong to the same criterion group are located on the same level on the y axis, while the x axis indicates the SRD values. The more ahead a criterion is located on the x axis (closer to 0), the more determinative it is. In this case, the most determinative criteria are the sub-criterion of 'Human Resources in Science and Technology, which defines the percentage of people employed in science and

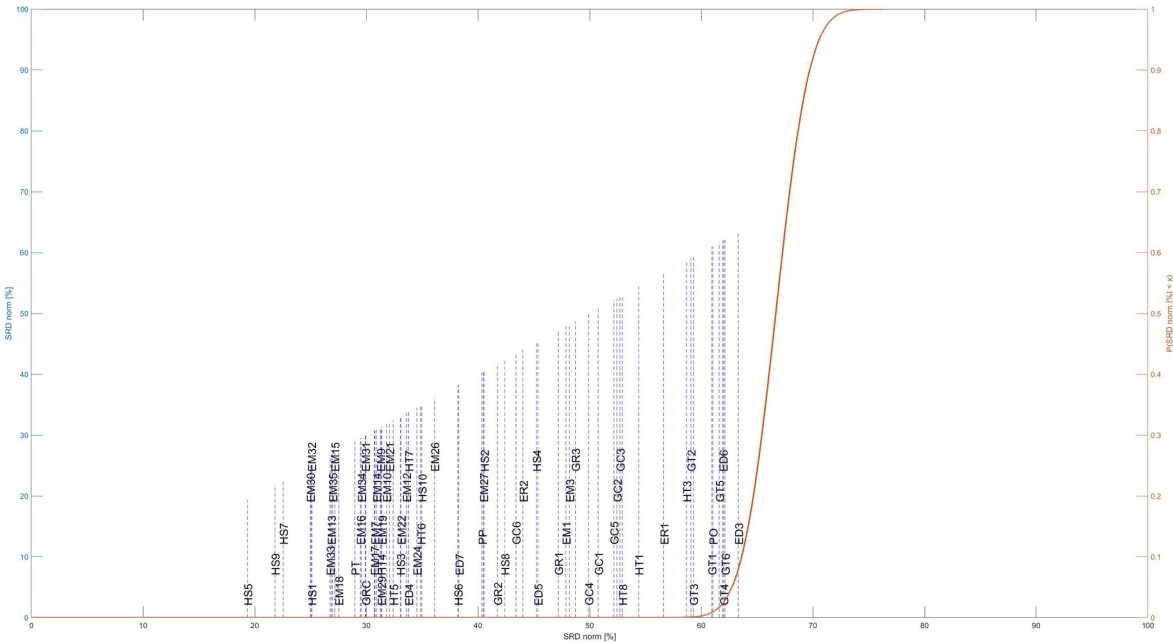

**Fig 9. The rank of criteria according to the P-SRD method.** The x axis shows the SRD values. Criteria belonging to the same criterion group are located on the same level on the y axis.

technology (HS5), and/or people in tertiary education (HS7) as well as scientists and engineers (HS9). This sub-criterion is closely followed by the sub-criteria of 'Employment rates of young people not in education and training' (EM), the Number of research-related institutions' (GRC), the 'Number of patent applications in the relevant field by regions' (PT), and 'Employment in technology and knowledge-intensive sectors' (HT).

It is notable, that the first criteria is ranked around the SRD value of 20, which means more suitable criteria (which rank closer to the reference) could be found to determine regional Industry 4.0 readiness.

## Discussion

The Promethee-based criterion net flow matrix **M** was analyzed using three suggested methods (see: Fig 1) to improve the Promethee-GAIA methodology as well as gain more information about the relationship between criteria and their determinative power concerning the measurement of regional I4.0 readiness. Each method identified the relationship between criteria from different perspectives, enabling the decision-maker to gain more complex information about the decision problem. The P-PFA clearly selected and grouped criteria into factors while considering and preserving the maximum amount of variance in criteria. It eliminates overly multicollinear and irrelevant criteria as well as clearly defines which group of criteria are correlated (belong) to the factor (see Fig 6), thereby improving its interpretation. The P-sPCA maximized the correlation between non-zero loadings and maintained the variances as high as possible. Furthermore, since the number of non-zero loadings was decreased, less criteria are involved in the same PCs. It can clearly be determined which criteria belong to which factor (see Fig 8). The P-SRD method ranked criteria and identified ones ranked in reverse (see Fig 9). The ranking differences provide information for the decision-maker about the similarities between criteria and their visualization enables the structure of the multicriteria decision problem to be analysed.

According to all three methods, the most determinative criteria concerning regional I4.0 readiness are 'Human Resources in Science and Technology' (HS) concerning people in tertiary education and/or employed in science and technology, 'Employment rates of young people not in education and training by sex, educational attainment level, years since completion of highest level of education and NUTS 2 regions' (EM), the 'Number of research-related institutions' (GRC) and the 'Number of I4.0-related news appearings' (GC).

Table 1 indicates the Promethee-ranked performance of the top-ten NUTS 2 regions with regard to the most determinative criteria. The criteria measure the percentage of the total population concerning people employed in science and technology (HS5), that is, scientists and engineers (HS9) as well as people studying in tertiary education and/or employed in science and technology (HS7). The analysed 314 regions perform within the following ranges according to the main criteria: HS5 3.3%—40.6%, HS9 0.5%—10.9%, HS7 8.6%—67.9%. Notably, Industry 4.0 developments are concentrated as only one or two regions finished top in the ranking from each leading country. The Promethee-based leader is the German Oberbayern region, followed by Berkshire, Buckinghamshire and Oxford region, then Stockholm. According to these three criteria, Stockholm performs the best in terms of HS5 and HS9 (highest percentage from all regions). Regarding HS7, the Warszawski Stołeczny region in Poland performed the best followed by Stockholm and Zürich.

The P-PFA method involved the most sub-criteria of 'Employment rates of young people not in education and training by sex, educational attainment level, years since completion of highest level of education and NUTS 2 regions' (EM).

**Table 1. Promethee-ranked performance of the top-ten NUTS 2 regions with regard to the most determinative criteria—'Human Resources in Science and Technology (Percentage of total population)' concerning people employed in science and technology (HS5), Scientists and engineers (HS9) and people studying in tertiary education and/or employed in science and technology (HS7).**

| Rank | NUTS 2 region | Name of the region | HS5 | HS9 | HS7 |
|------|---------------|--------------------|-----|-----|-----|
| 1 | DE21 | Oberbayern | 33.4 | 8.4 | 47.2 |
| 2 | UKJ1 | Berkshire, Buckinghamshire, and Oxfordshire | 32.2 | 9.7 | 53.6 |
| 3 | SE11 | Stockholm | 40.6 | 10.9 | 56.8 |
| 4 | NL32 | North Holland | 32.7 | 7.5 | 46.8 |
| 5 | CH04 | Zurich | 37.7 | 10.5 | 56.8 |
| 6 | SE23 | West Sweden | 31.6 | 8.1 | 45.3 |
| 7 | PL91 | Warszawski stołeczny | 39.2 | 9.8 | 57 |
| 8 | UKK1 | Gloucestershire, Wiltshire and Bristol/Bath area | 27.8 | 8.3 | 46.1 |
| 9 | NL41 | North Brabant | 27.1 | 6.8 | 39 |
| 10 | NO01 | Oslo og Akershus | 36.7 | 10.5 | 55.2 |

The P-sPCA involved additional criteria related to lifelong learning activities (ED5–7) and the 'Number of graduates in Engineering, Manufacturing and Construction (BSc, MSc, PhD)' (GR3).

The P-SRD also ranked the criterion of 'Number of patent applications' (PT), moreover, identified criteria ranked in reverse (ED6 and HT3), which were still included among the PCs defined by the P-sPCA method after these criteria had been reversed. P-SRD also revealed that criteria selected based on the P-PFA method cover a broader scale with regard to the SRD values (20–55), while the P-sPCA-selected criteria were located within a narrower SRD range (35–55). Criteria loaded into the two PCs based on P-sPCA are quite similar to each other.

Fig 10 indicates a Promethee-based rank performed on the criteria selected by P-PFA. The value of loadings indicates two criteria groups which have been shown if Figs 6 and 7. In sub-

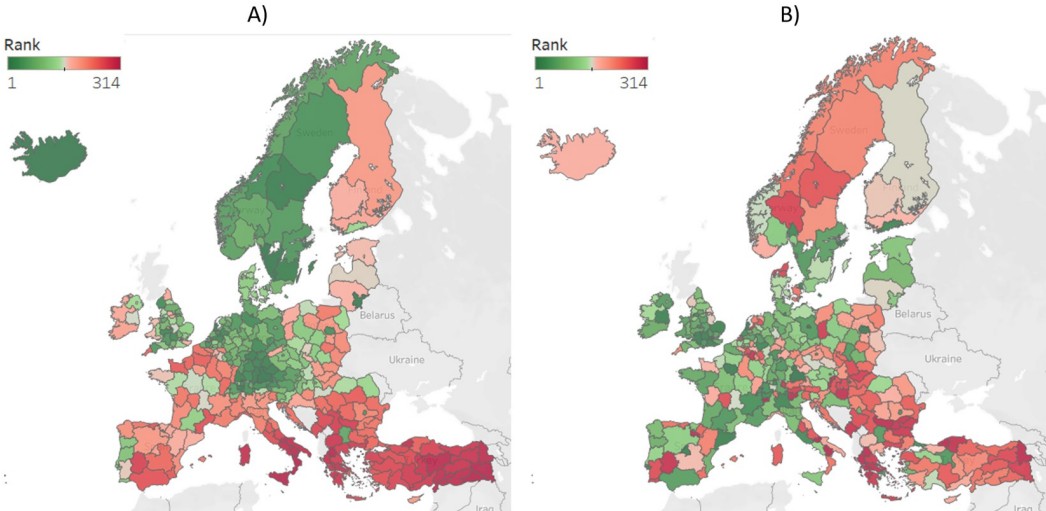

**Fig 10. Map visualization of the Promethee-based I4.0 readiness ranking of European NUTS 2 regions based on the two loadings of P-PFA.** It provides a visual representation of how regions perform/rank considering the two major criterion groups. Sub-figure A) represents regional rank based on the first loading, which includes criteria of employment rates and job opportunities. Sub-figure B) represents regional rank based on the first loading, which includes criteria of academia sector and its employees in R&D fields and I4.0-related news.

figure A) regions are ranked according to the criteria loaded into the first factor, while sub-figure B) represents the rank of regions based on the criteria loaded into the second factor. The visualization of regional rank based on the separation of two factors indicates which criteria groups (A—employment rates and job opportunities; B—employment in R&D and the I4.0-related news) regions have to develop. The German Oberbayern region performed the best on both factors, such as Central European regions seems to be good on both factors. While the regions of Iceland, Sweden, and Norway seem to have higher employment rates and job opportunities. Still, the employment may not come from the science and technology sectors, and public awareness measured through I4.0-related news seems to be lower. On the contrary, for example, some regions of France (e.g., regions in Languedoc-Roussillon-Midi-Pyrénées) and Spain (e.g., Cento regions), or West Finland may have lower employment rates. Still, the number of employees in the academia sector seems to be higher along with public awareness of Industry 4.0. However, some regions perform poorly on both factors, such as regions of Italy (e.g., South and Northwest Italy) and Turkey (e.g., Central and Northeast Anatolia Regions and the West Black Sea Regions).

## Conclusions

This paper introduced three methods to improve the interpretation of Promethee-GAIA-based decision-making. Both the Principal factoring with rotation and communality analysis (P-PFA) and the Sparse PCA method (P-sPCA) algorithms carefully select the important criteria.

The applicability of the methods is presented through an application study on measuring the regional Industry 4.0 readiness of NUTS 2-classified regions. The analysis of the Promethee flow matrix can provide information about the similarities between the criteria to gain more of an insight into multicriteria decision problems.

Table 2 summarizes the most advantageous features of the methods and their utilization potential. P-PFA is efficient in terms of selecting criteria and classifying them into factors. P-PFA eliminates criteria based on Measure of Sampling Adequacy (MSA) and communality values as well as preserves the maximum amount of variance. It is efficient in terms of quick screening, and when fewer criteria are needed for our model. The P-sPCA selects only a few relevant variables and classifies them into two PCs. This is useful if no visualization is needed but a compact composite criterion including the most determinative criteria is preferred. The Sum of Ranking Differences method (P-SRD) is presented as a special case of the Promehtee

**Table 2. Summary of the advantages and utilization potential of the proposed methods.**

| Methods | Most advantageous features | Utilization potential |
|---------|----------------------------|----------------------|
| P-PFA | Selects and classifies criteria | In the case of many redundant criteria |
|  | Preserves the maximum amount in variance of criteria | Only a few criteria are needed for our model |
|  |  | Needs a quick screening function |
| P-sPCA | Classifies criteria | No need for visualization |
|  | Creates independent composite criteria | Needs a composite criterion that only includes a few of the most determinative criteria |
| P-SRD | Visualizes similarities between and groupings of criteria in terms of Promethee | Dedicated criteria rank for the Promethee-generated rank |
|  | Visualizes the SRD probability function |  |
|  | Defines the rank of criteria based on the distances from the reference |  |
|  | Provides a statistical test |  |

method. SRD is an efficient tool to visualize and evaluate similarities between criteria based on the sum of ranking differences according to a golden standard (in this case, the average). Therefore, a dedicated rank of criteria with regard to the Promethee-based rank can be made. Furthermore, it enables the comparison of the results of the other methods to be compared and identifies similarities between criterion groups. The rank of criteria enables how determinative a criterion is to be defined and provides a statistical test.

## Supporting information

**S1 Appendix. Criteria description.** This file provides the description of criteria included in the analysis.
(XLSX)

## Author Contributions

**Conceptualization:** János Abonyi, Zsolt T. Kosztyán, Károly Héberger.

**Formal analysis:** János Abonyi, Tímea Czvetkó, Zsolt T. Kosztyán, Károly Héberger.

**Funding acquisition:** János Abonyi, Károly Héberger.

**Methodology:** János Abonyi, Zsolt T. Kosztyán, Károly Héberger.

**Supervision:** János Abonyi, Károly Héberger.

**Validation:** János Abonyi, Tímea Czvetkó, Zsolt T. Kosztyán, Károly Héberger.

**Visualization:** János Abonyi, Tímea Czvetkó, Zsolt T. Kosztyán.

**Writing – original draft:** János Abonyi, Tímea Czvetkó, Zsolt T. Kosztyán.

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
