## [Decision Letter · Decision Letter 0]

16 Dec 2021

PONE-D-21-28324Factor analysis, Sparse PCA, and Sum of Ranking Differences-based improvements of the Promethee-GAIA multicriteria decision support techniquePLOS ONE

Dear Dr. Abonyi,

Thank you for submitting your manuscript to PLOS ONE. After careful consideration, we feel that it has merit but does not fully meet PLOS ONE’s publication criteria as it currently stands. Therefore, we invite you to submit a revised version of the manuscript that addresses the points raised during the review process.

We look forward to receiving your revised manuscript.

Kind regards,

Fausto Cavallaro, PhD

Academic Editor

PLOS ONE

Reviewers' comments:

Reviewer's Responses to Questions

**Comments to the Author**

1. Is the manuscript technically sound, and do the data support the conclusions?

Reviewer #1: Yes

Reviewer #2: Yes

Reviewer #3: Yes

2. Has the statistical analysis been performed appropriately and rigorously? 

Reviewer #1: Yes

Reviewer #2: Yes

Reviewer #3: Yes

3. Have the authors made all data underlying the findings in their manuscript fully available?

Reviewer #1: Yes

Reviewer #2: Yes

Reviewer #3: Yes

4. Is the manuscript presented in an intelligible fashion and written in standard English?

Reviewer #1: Yes

Reviewer #2: Yes

Reviewer #3: Yes

5. Review Comments to the Author

Reviewer #1: The paper "Factor analysis, Sparse PCA, and Sum of Ranking Differences-based improvements of the Promethee-GAIA multicriteria decision support technique" represents a very good study with strong and novel proposed methodology.

The authors are very familiar with the fields covered in the paper. It is well written with almost all necessary elements. The authors have proposed a lot of details in order to verify their results.

The paper has great potential and can be accepted after major corrections:

- A good previous study overview is missing.

- In the introduction section you gave an overview of other relevant studies, and try to show the link between previous studies and your paper. That is partially good, but the following tasks should be fulfilled also:

the introduction should give an overview of the field significance, and should consider the following main questions: What are the gaps in literature? What are the contributions of this study? What are the main aims of this article?"

- Related to the previous comment. You have given a short overview of the previous study in the introduction, but not in the proper way. You didn't explain the most of studies, you have only listed studies.

- Most of the current section Introduction should be moved to new-formed 2. Literature review

- Figures shouldn't be part of introduction.

- You should include the following references related to application study:

Widjajanto, S., Purba, H. H., & Jaqin, S. C. (2020). Novel POKA-YOKE approaching toward industry-4.0: A literature review. Operational Research in Engineering Sciences: Theory and Applications, 3(3), 65-83.

Chakraborty, S., Chattopadhyay, R., & Chakraborty, S. (2020). An integrated D-MARCOS method for supplier selection in an iron and steel industry. Decision Making: Applications in Management and Engineering, 3(2), 49-69.

Reviewer #2: This study uses the Promethee-GAIA method as a multi-criteria decision support technique that defines the aggregate ranking of several criteria and visualizes it based on Principal Component Analysis (PCA). The author is quite good at presenting analytical data related to the topic under study.

Reviewer #3: The paper has interesting topics and is in accordance with the primary goal stated by the authors - supporting PROMETHEE-based decision-making by integrating various statistical methods in standard PROMETHEE procedure, that enable better identification and interpretation of relationships between criteria. The PROMETHEE-GAIA algorithm is well presented as well as procedure of integration with P-PFA, P-sPCA and P-SRD. This part of the paper can be improved by more detailed elaboration of PCA integrations in MCDM methods. Some of useful titles to cite could be Dugger et al (2021) Principal Component Analysis in MCDM: An exercise in pilot selection, Expert Systems with Applications, Volume 188, February 2022, 115984, https://doi.org/10.1016/j.eswa.2021.115984; Ranđelovic et al. Weight coefficients determination based on parameters in factor analysis, Metalurgia International, XVIII(3), str. 128-132, https://www.researchgate.net/publication/298834442_Weight_coefficents_determination_based_on_parameters_in_factor_analysis ; Ravoudabadi et al. An integrated weighting and ranking model based on entropy, DEA and PCA considering two aggregation approaches for resilient supplier selection problem, Journal of Computational Science, Vol. 40, 2020, 101074,https://doi.org/10.1016/j.jocs.2019.101074, etc.

The second part of the paper, related to application of proposed methodology in industry 4.0 assessment across European NUTS2 regions need to be extended and better explained. Although, authors have good applied results, strong results discussion is missing. Results visualisation will provide clear spatial presentation of industry 4.0 development in Europe and in the same time make the paper more attractive to the readers. The strong background of the similar applications of PROMETHEE-GAIA (both integrated with other procedures or applied in the composite indicators creation) is needed to be presented. Please see Greco et al (2021) The ordinal input for cardinal output approach of non-compensatory composite indicators: the PROMETHEE scoring method, European Journal of Operational Research, 288(1):225-246, https://doi.org/10.1016/j.ejor.2020.05.036; Stankovic et al (2021) An integrated approach of PCA and PROMETHEE in spatial assessment of circular economy indicators, Waste Management, 128: 154-166, https://doi.org/10.1016/j.wasman.2021.04.057; etc. Finally some results on other research related on industry 4.0 analyses can be presented as results discussion.

6. PLOS authors have the option to publish the peer review history of their article (what does this mean?). If published, this will include your full peer review and any attached files.

Reviewer #1: No

Reviewer #2: No

Reviewer #3: No

---

## [Author Response · Author response to Decision Letter 0]

11 Jan 2022

Reply to editor

Title: Factor analysis, Sparse PCA, and Sum of Ranking Differences-based improvements of the Promethee-GAIA multicriteria decision support technique

Authors: János Abonyi, Tímea Czvetkó, Zsolt T. Kosztyán, Károly Héberger

Ref. no.: PONE-D-21-28324

Dear Editor,

First, we would like to thank you for the opportunity to revise our manuscript. We revised our manuscript according to the Reviewers’ remarks. The section of the introduction was extended with relevant literature, MCDM application examples and a new subsection was formed called the The proposed multicriteria decision support framework. All the suggested studies were referred to, and the most related works were studied in detail. We addressed all the comments and suggestions as explained below. We hope that the revised manuscript meets both your and the reviewers’ expectations. The changes in the manuscript highlighted in blue.

Yours Sincerely,

The Authors

 

Response to reviewers

Dear Reviewer #1,

We would like to thank you for the opportunity to revise our manuscript. According to your remarks, we revised our manuscript. The section of the introduction is extended and a new section entitled The proposed multicriteria decision support framework is formed. All the suggested studies are referred to, and the most related works were more studied. We highlighted the research gap and the main contribution to the literature. In addition, new map visualization in the Discussion section highlights the result of the proposed method. 

We hope, that the revised manuscript meets both your and the reviewers’ expectations.

Yours Sincerely,

The Authors

Reviewer #1: The paper "Factor analysis, Sparse PCA, and Sum of Ranking Differences-based improvements of the Promethee-GAIA multicriteria decision support technique" represents a very good study with strong and novel proposed methodology.

The authors are very familiar with the fields covered in the paper. It is well written with almost all necessary elements. The authors have proposed a lot of details in order to verify their results.

The paper has great potential and can be accepted after major corrections:

Thank you for your positive and unambiguous evaluation.

- A good previous study overview is missing.

Thank you for your valuable remark. It has revealed that a better review of recent analysis and trends is missing. The introduction section is extended to study numerous related works and analyzes the present trends and state of the art.

 - In the introduction section you gave an overview of other relevant studies, and try to show the link between previous studies and your paper. That is partially good, but the following tasks should be fulfilled also: the introduction should give an overview of the field significance, and should consider the following main questions: What are the gaps in literature? What are the contributions of this study? What are the main aims of this article?"

Thank you for your valuable comment. The revised Introduction section emphasizes the research gap, the aim of the article, and our contribution to the literature. The more detailed aim of the framework is discussed in a new subsection entitled The proposed multicriteria decision support framework.

- Related to the previous comment. You have given a short overview of the previous study in the introduction, but not in the proper way. You didn't explain the most of studies, you have only listed studies.

Thank you for your comment. We admit that the Introduction section was too short and aimed exclusively at methodological development. The applications had served as an illustration only. The Introduction section has been extended with recent literature to analyze trends, state of the art, and applications.

 - Most of the current section Introduction should be moved to new-formed 2. Literature review

Thank you for your comment. The suggested Literature review part has been added to the Introduction section since the journal requires only three major sections: Introduction, Materials and Methods, and Results, Discussion, Conclusion (the last one can be separated).

Please see under the sections of Manuscript Organization and Parts of Submission: https://journals.plos.org/plosone/s/submission-guidelines#loc-manuscript-organization

- Figures shouldn't be part of introduction.

Thank you for your suggestion. The Introduction section has been modified: a literature review part is added, while all figures and related paragraphs are moved from the introduction to the following new subsection: The proposed multicriteria decision support framework.

- You should include the following references related to application study:

1) Widjajanto, S., Purba, H. H., & Jaqin, S. C. (2020). Novel POKA-YOKE approaching toward industry-4.0: A literature review. Operational Research in Engineering Sciences: Theory and Applications, 3(3), 65-83.

2) Chakraborty, S., Chattopadhyay, R., & Chakraborty, S. (2020). An integrated D-MARCOS method for supplier selection in an iron and steel industry. Decision Making: Applications in Management and Engineering, 3(2), 49-69.

Thank you for your valuable comment. The suggested literature revealed that application examples are missing from the manuscript. A paragraph has been added to the Application study subsection, in which the importance of multicriteria decision-support methods in terms of Industry 4.0 development and strategic planning is emphasized. The suggested articles have been implemented in this paragraph.

 

Dear Reviewer #2,

We would like to thank you for the opportunity to revise our manuscript. According to your remarks, we revised our manuscript. The section of the introduction is extended and a new subsection entitled The proposed multicriteria decision support framework is formed. We highlighted the research gap and the main contribution to the literature. In addition, new map visualization in the Discussion section highlights the result of the proposed method. 

We hope, that the revised manuscript meets both your and the reviewers’ expectations.

Yours Sincerely,

The Authors

Reviewer #2: This study uses the Promethee-GAIA method as a multi-criteria decision support technique that defines the aggregate ranking of several criteria and visualizes it based on Principal Component Analysis (PCA). The author is quite good at presenting analytical data related to the topic under study.

Thank you for your positive and unambiguous evaluation.

 

Dear Reviewer #3,

We would like to thank you for the opportunity to revise our manuscript. All the suggested studies are cited, and the most related works were studied in detail. New maps and figures highlight the result of the proposed method. In addition, in the revised paper, the section of the introduction is extended and a new subsection entitled The proposed multicriteria decision support framework is formed. In the section of Discussion, the relevancy and importance of utilizing MCDM methods in Industry 4.0 development is highlighted. We also highlighted the research gap and the main contribution to the literature.

We hope, that the revised manuscript meets both your and the reviewers’ expectations.

Yours Sincerely,

The Authors

Reviewer #3: The paper has interesting topics and is in accordance with the primary goal stated by the authors - supporting PROMETHEE-based decision-making by integrating various statistical methods in standard PROMETHEE procedure, that enable better identification and interpretation of relationships between criteria. The PROMETHEE-GAIA algorithm is well presented as well as procedure of integration with P-PFA, P-sPCA and P-SRD. 

Thank you for your positive and unambiguous evaluation.

This part of the paper can be improved by more detailed elaboration of PCA integrations in MCDM methods. Some of useful titles to cite could be 

Thank you for the valuable list of references. All of them are cited in the revised manuscript. In the extended introduction we discuss that the shortcomings of these works are the same as the Promethee - Gaia as they use PCA instead of more advanced dimension reduction methods.

3) Dugger et al (2021) Principal Component Analysis in MCDM: An exercise in pilot selection, Expert Systems with Applications, Volume 188, February 2022, 115984, https://doi.org/10.1016/j.eswa.2021.115984;

It is cited in the section of the introduction. This paper uses only PCA in MCDM.

4) Ranđelovic et al. Weight coefficients determination based on parameters in factor analysis, Metalurgia International, XVIII(3), str. 128-132, https://www.researchgate.net/publication/298834442_Weight_coefficents_determination_based_on_parameters_in_factor_analysis ; 

This paper has been cited in the revised manuscript. We think this work is closer to our scope of the study. Unfortunately, this study could not be reached until our submission. The integration of AHP and FA, where FA provides the weights to AHP is relevant. 

5) Ravoudabadi et al. An integrated weighting and ranking model based on entropy, DEA and PCA considering two aggregation approaches for resilient supplier selection problem, Journal of Computational Science, Vol. 40, 2020, 101074,

https://doi.org/10.1016/j.jocs.2019.101074 , etc.

We also cited this study. Integrating DEA and PCA is relevant, however, because of the shortcomings of PCA the drawbacks are the same as the 3).

The second part of the paper, related to application of proposed methodology in industry 4.0 assessment across European NUTS2 regions need to be extended and better explained. 

Thank you for your valuable comment. Figure 9 is added to the Discussion section, which indicates a map visualization of regional ranks regarding values of criteria with explanations. 

Although, authors have good applied results, strong results discussion is missing. 

Thank you for your comment. We improved the section of the discussion.

Results visualisation will provide clear spatial presentation of industry 4.0 development in Europe and in the same time make the paper more attractive to the readers. 

The strong background of the similar applications of PROMETHEE-GAIA (both integrated with other procedures or applied in the composite indicators creation) is needed to be presented. 

Please see 

6) Greco et al (2021) The ordinal input for cardinal output approach of non-compensatory composite indicators: the PROMETHEE scoring method, European Journal of Operational Research, 288(1):225-246, https://doi.org/10.1016/j.ejor.2020.05.036;

7) Stankovic et al (2021) An integrated approach of PCA and PROMETHEE in spatial assessment of circular economy indicators, Waste Management, 128: 154-166, https://doi.org/10.1016/j.wasman.2021.04.057 ; etc. 

Thank you for the list of papers. All have been added to the Introduction section along with other literature giving more details and application examples of the MCDM methods.

Finally, some results on other research related on industry 4.0 analyses can be presented as results discussion.

Thank you for your valuable remark. It has revealed that more explanation is needed regarding the connection between Industry 4.0 and MCDM techniques and their application examples. A paragraph in the Discussion section has been added, which tackles the aspects of Industry 4.0 readiness measurements and their significance.

---

## [Decision Letter · Decision Letter 1]

8 Feb 2022

Factor analysis, Sparse PCA, and Sum of Ranking Differences-based improvements of the Promethee-GAIA multicriteria decision support technique

PONE-D-21-28324R1

Dear Dr. Abonyi,

We’re pleased to inform you that your manuscript has been judged scientifically suitable for publication and will be formally accepted for publication once it meets all outstanding technical requirements.

Kind regards,

Fausto Cavallaro, PhD

Academic Editor

PLOS ONE

Reviewers' comments:

Reviewer's Responses to Questions

**Comments to the Author**

1. If the authors have adequately addressed your comments raised in a previous round of review and you feel that this manuscript is now acceptable for publication, you may indicate that here to bypass the “Comments to the Author” section, enter your conflict of interest statement in the “Confidential to Editor” section, and submit your "Accept" recommendation.

Reviewer #1: All comments have been addressed

Reviewer #3: All comments have been addressed

2. Is the manuscript technically sound, and do the data support the conclusions?

Reviewer #1: Yes

Reviewer #3: Yes

3. Has the statistical analysis been performed appropriately and rigorously? 

Reviewer #1: Yes

Reviewer #3: Yes

4. Have the authors made all data underlying the findings in their manuscript fully available?

Reviewer #1: Yes

Reviewer #3: Yes

5. Is the manuscript presented in an intelligible fashion and written in standard English?

Reviewer #1: Yes

Reviewer #3: Yes

6. Review Comments to the Author

Reviewer #1: The authors have improved quality of the paper and now can be accepted for publication. My comments are properly addressed.

Reviewer #3: After changes, the paper is acceptable for publication. The authors have taken into account all the remarks and suggestions, so now the work is of significantly improved quality.

---

## [Editor Report · Acceptance letter]

10 Feb 2022

PONE-D-21-28324R1 

Factor analysis, Sparse PCA, and Sum of Ranking Differences-based improvements of the Promethee-GAIA multicriteria decision support technique 

Dear Dr. Abonyi:

I'm pleased to inform you that your manuscript has been deemed suitable for publication in PLOS ONE. Congratulations! Your manuscript is now with our production department. 

Kind regards, 

on behalf of

Professor Fausto Cavallaro 

Academic Editor

PLOS ONE